# Study on water resources carrying capacity based on Pressure-State-Response modeling: An empirical study of the urban agglomeration in Central Yunnan, China

**Jing Zhang[1], Wenchuang Guan[2,3], Guangping Wu[2], Jing Wang[2]\*, Biyu Rao[2], Bulin Zhang[1]**

1 School of Business Administration, Liaoning Technical University, Huludao, China, 2 School of Architecture and Engineering, Yunnan Agricultural University, Kunming, China, 3 School of Architecture and Engineering, Henan Vocational University of Science and Technology, Zhoukou, China

\* 471910083@stu.lntu.edu.cn

## Abstract

Water resources carrying capacity (WRCC) is the basis for sustainable regional development and an important indicator of core competitiveness, and its quantitative assessment and comparison is a key link in clarifying the development capacity of the region. The study comprehensively considers economic, social, resource, environmental and ecological factors, constructs a WRCC evaluation index system based on the Pressure-State-Response (PSR) model, adopts the entropy value method to assign weights to each index, and utilizes the set-pair analysis method and the obstacle diagnostic model to evaluate WRCC of the urban agglomerations in central Yunnan (UACY) in the period from 2008 to 2020. The results show that the comprehensive development of WRCC of UACY is characterized by stage-by-stage evolution in the time dimension, with a decreasing trend in the carrying capacity from 2008 to 2012, and an overall fluctuating upward development trend from 2013 to 2020. In the spatial dimension, Kunming's WRCC is generally poor, and Honghe and Yuxi have the greatest advantages in water resources storage and conservation capacity. The stress of water use in Kunming is higher, but Kunming has advantages in industrial structure and water resources development and utilization rate. Through the diagnosis of obstacle degree, the main obstacle factors of WRCC have large differences among cities (states), but the main subsystems constraining WRCC are all pressure subsystems. The results of the study can provide data support for water resources related policies and rational water resources dispatching in the UACY.

## Introduction

Water resources are irreplaceable strategic resources, and although different forms of cyclic transformation can be realized, the continuous exploitation and waste of human beings as well as the generation of polluting behaviors have gradually reduced the amount of available water

**Data Availability Statement:** All relevant data are within the manuscript and its Supporting information files.

**Funding:** This study was funded by the National Natural Science Foundation of China (No. 54322066027). The funders had no role in study design, data collection and analysis, decision to publish, or preparation of the manuscript.

**Competing interests:** The authors have declared that no competing interests exist.

resources [1–3]. The irrational exploitation of water resources and the lack of scientific protection will ultimately sound the alarm of water resources, constrain human production and life, and hinder economic development and social progress [4, 5]. Carrying capacity is initially a kinetic physical concept that reflects the maximum bearing capacity of an object before it is destroyed. After the 1960s, anthropologists and biologists used the concept of carrying capacity in human ecology to describe the maximum tolerance of a regional system to the external environment. The concept and meaning of carrying capacity has since changed from a physical concept to one that reflects the limits of the environment or ecosystem to carry development and specific activities. The WRCC can be defined as the ability of a region's resources to support the sustainable development of its economic, social and ecological systems under certain conditions [6], which is a composite system involving economic, social, ecological environment and other systems [7], and is an important indicator to measure the coordination between water resources and the social economy as well as the ecological environment, which can reflect the supportive capacity of water resources in the region [8]. In 2020, the People's Government of Yunnan Province issued the Development Plan for the UACY, which proposed that it will build a spatial pattern and modern industrial system adapted to the carrying capacity of resources and environment, and the interconnected infrastructure network will be improved. By 2025, the functions of the UACY will be basically improved, gathering 50% of the population and 68% of the regional GDP. As the most economically developed region in Yunnan Province, the UACY will increase the intensity of consumption of water resources in the process of future development, as well as pollute the water environment and affect WRCC. Therefore, in order to explore the coordinated relationship between the economy, society and water resources in the UACY, and to scientifically and reasonably develop, utilize and protect the water resources in the UACY, the first and foremost element is to evaluate the carrying capacity of the existing water resources.

Due to the fact that WRCC involves more factors and a wide range of levels [9], many scholars have used different methods to evaluate WRCC from different aspects, and have achieved fruitful research results. WRCC evaluation research is mainly reflected in the construction of the evaluation index system, the determination of index weight and the selection of evaluation model. Currently, there are many applied evaluation methods of WRCC, including hierarchical analysis method, principal component analysis method, fuzzy comprehensive evaluation method, TOPSIS method, system dynamics method and set-pair analysis method [10–16] [10–16], and most of them use the TOPSIS model, DPSIR model and PSR model [17–19] to conduct the analysis and research. For example, Zhang et al. [20] used a system dynamics approach to study the WRCC of the Beijing-Tianjin-Hebei region, and concluded that in the resource-scarce Beijing-Tianjin-Hebei region, it is necessary to improve the efficiency of water resources utilization, pay attention to the development of water resources and the protection of the water environment in order to achieve the goal of improving the WRCC. Liu et al. [21] constructed a WRCC evaluation model based on the PSR model to evaluate the WRCC of the Yangtze River Delta, and proposed countermeasures for maintaining the balance between regional water resources supply and demand. Tian et al. [22] applied the variable-weight TOPSIS method to study the WRCC of nine provinces and two cities in the Yangtze River Economic Belt, and concluded that the WRCC of the Yangtze River Economic Belt is greatly influenced by the regional water resources endowment status and ecological and environmental conditions. Yang et al. [23] studied the water resources carrying condition of the Manas River Basin from the time dimension through the fuzzy comprehensive evaluation method. Tong et al. [24] used principal component analysis to study the WRCC of Nanjing, and concluded that population and economic development, water resources endowment, and resource use efficiency are the main factors affecting the WRCC in Nanjing. Tang et al. [25] used set-

pair analysis to diagnose and identify the evolutionary characteristics and vulnerability indicators of WRCC in Guiyang City from 2005 to 2017. Wang et al. [26] used fuzzy set-pair analysis and obstacle diagnostic model to evaluate the WRCC of each province in the Yellow River Basin from 2008 to 2017 and identified and analyzed the barrier factors of WRCC.

In summary, substantial progress has been made in both the theoretical and practical aspects of WRCC assessment. The assessment research system has been initially formed, which has been widely applied in urban agglomerations and watersheds in developed regions, which is of certain significance, but there are still limitations. Some of these are that (1) In the selection of indicators, the whole process and internal mechanism of the evolution and change of the WRCC are neglected, and there is a lack of systematic linkage. (2) In terms of weight determination, most of them are determined through expert consultation. Due to the expert experience and their way of thinking, their subjectivity is strong and the use of objective information is greatly weakened [27]. (3) At this stage, the research results mainly focus on regions such as urban agglomerations and watersheds in developed areas [28–30], and there is a lack of research on the WRCC of the southwestern border zone, especially for the WRCC of the plateau basin is less. The Southwest China Plateau region suffers from resource water scarcity caused by mountainous terrain blocking water flow and engineering water scarcity caused by rapid urban development. How to construct a WRCC evaluation model that can reflect the relationship between urban social and economic development and the internal evolution mechanism of WRCC, and put forward an objective evaluation method is an important prerequisite for the development, utilization, protection and management of urban water resources in southwest Plateau.

In view of the above challenges and requirements, this study focuses on the UACY, where resource water scarcity and engineering water scarcity co-exist. By analyzing the correlation between the economy, society, environment and WRCC of the UACY on the southwestern plateau, this study constructed a WRCC evaluation index system based on the PSR model that can reflect the relationship between urban socio-economic development and the internal evolution mechanism of the WRCC, and put forward a set-pair analysis to evaluate the WRCC of the UACY in the period of 2008–2020, and identified the main obstacles to the WRCC. The study clarifies the main reasons for the resource and engineering water shortage problems faced by the water resources of the UACY, and puts forward countermeasures to promote the coordinated development of water resources and socio-economics accordingly. The results of the study provide references for regional planning and water resources planning, as well as a basis for decision-making on the rational development and utilization of water resources.

## Research area and research methods

### Overview of the study area

The urban agglomerations in central Yunnan (100°43'-104°49'E longitude, 22°59'-27°3'N latitude) is located in the east-central part of Yunnan Province, and its scope includes Kunming City, Qujing City, Yuxi City, Chuxiong Prefecture, and the northern part of Honghe Prefecture, and the specific location of the study area is shown in Fig 1. Municipal boundary [31] were obtained from the Resource and Environment Science Data Center of the Chinese Academy of Sciences (http://www.resdc.cn/). The UACY has a complex topographical structure, with mountain ranges oriented from northwest to southeast, and the terrain is high in the north and low in the south, and most of the area is between 1,500 and 2,800m above sea level, with high altitude and large elevation difference, with more mountainous land and less flat land, and it is the core area of the Radiation Centre facing Southeast Asia and South Asia. With a land area of 111,400 square kilometers, a GDP of 1.02 trillion Yuan and a resident

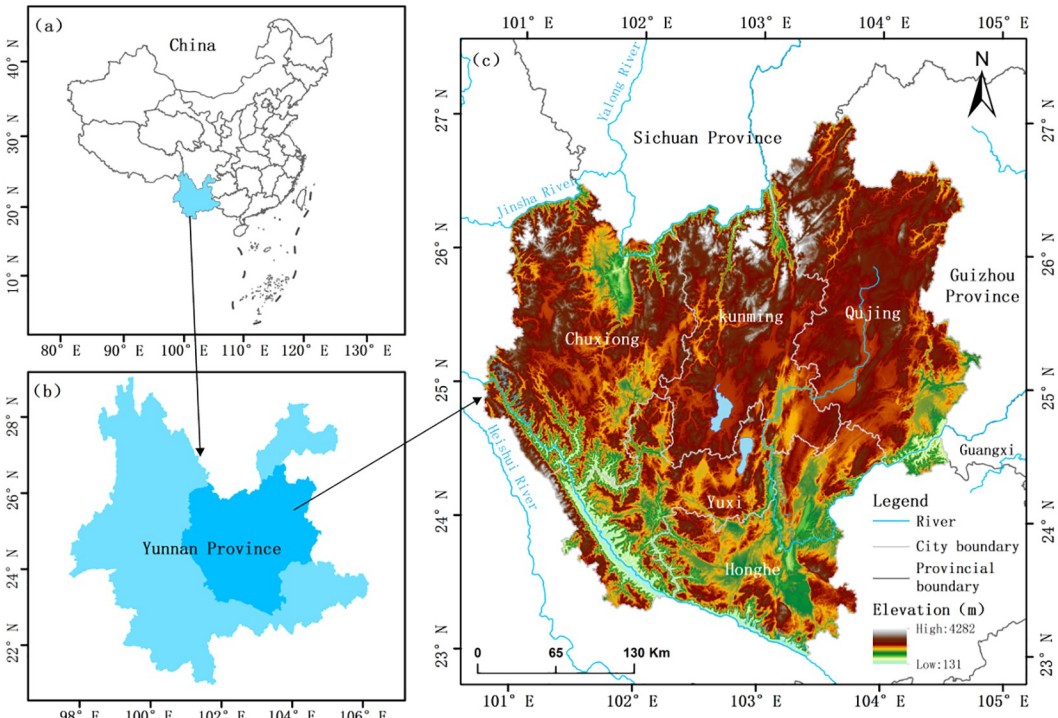

**Fig 1. Location of the study area.** Note: Map created using ArcGIS [10.2]. Data downloaded from the Resource and Environment Science Data Center of the Chinese Academy of Sciences (http://www.resdc.cn/).

population of 21.27 million at the end of 2018, accounting for 28.3%, 61.6% and 44.1% of the province, respectively, the UACY has the strongest economic strength, a better development status quo, and the greatest development potential, but it is the region with the most serious water shortage in Yunnan Province. The region's water resources holdings, population and industrial development are extremely uncoordinated, and water resource shortages have become the biggest "bottleneck" constraining the population and industrial development of the UACY. Coupled with the irrational development and utilization of water resources and the lack of water resource protection, most counties and districts in the UACY are experiencing deteriorating water environments, a continuous lack of water resources, and a lack of improvement in the level of water resource management. According to statistics, in 2020, the available water resources in the UACY only account for 15.3% of the total water resources in Yunnan Province, and there is a co-existence of resource shortage and engineering shortage in the region, with the problem of cities crowding out the water in the rural areas and production crowding out the water in the ecological environment becoming more and more prominent. The water resources situation is characterized by uneven spatial distribution, lack of water, and large yearly changes, and the contradiction between supply and demand is becoming more and more prominent.

## Data source

The data used in this study come from "China Urban Statistical Yearbook", "China County Statistical Yearbook", "Yunnan Provincial Statistical Yearbook", "Yunnan Provincial Water Resources Bulletin", "Yunnan Provincial Ecological and Environmental Condition Bulletin", and "Statistical Bulletin of National Economic and Social Development of Various

Prefectures". Some of the indicator data were obtained by relevant calculations, and the missing data for some years were obtained by linear interpolation.

## Research methodology

**Construction of WRCC evaluation indicator system based on Pressure-State-Response model.** PSR model is put forward by Canadian statisticians. The theory of the model is that human beings obtain resources from the natural environment, and after social and economic production activities, pressure is exerted on the environment, thus changing the state of the resources, and human beings continuously make adjustments to the resources and production activities, forming the relationship of "pressure-state-response", that is, the PSR model [32, 33]. The three systems influence and constrain each other, forming a dynamic impact and assessment system of water resources carrying system [34]. In the PSR model, pressure subsystem refers to the impact of human activities on the carrying load of water resources, so human activities related to water use are included as indicators for evaluating the carrying pressure. State subsystem refers to the current level of water resources under the pressure of the water resources environment. Response subsystem refers to the transformation and connection between the water resources environment and human activities [35].

WRCC is affected by various aspects and involves many factors, and the selection of evaluation indicators is the key to the evaluation of WRCC [36]. From the analysis of PSR model, it can be seen that human activities are the main influencing factors of WRCC, which are mainly reflected in the three aspects of urban society, economy and environment. Pressure indicators should reflect the effect of socio-economic-environmental impacts on the WRCC system; state indicators should reflect the changes in the state of the water resources system in the socio-economic-environmental context, taking into account the relationship of sustained and coordinated development of the various states; and response indicators should reflect the measures taken or policies enacted to safeguard water resources and the environment. Therefore, in order to ensure the construction of a reasonable evaluation index system, this study is based on the construction of the PSR framework model based on the consideration of urban social, economic and environmental factors, and combined with the previous WRCC of the relevant studies, following the principles of representativeness, accessibility, systematicity and operability of the indicator data [37], to establish the evaluation index system of the WRCC in three aspects, as shown in Table 1. According to the relationship between the intensity value of WRCC and the value of the evaluation index, the evaluation index of WRCC is divided into positive and negative indexes, the positive index is the larger the value of the index the stronger the WRCC, and the negative index is the smaller the value of the index the stronger the WRCC.

**Determination of indicator weights.** *(1) Raw data standardization.* It is necessary to judge the type to which the indicator belongs before calculation. In order to avoid the influence of different scale of each indicator, this paper adopts the method of standardization of polar deviation to standardize each evaluation indicator [38, 39].

Standardized formula for positive indicators:

$$b = (X - X_{min})/(X_{max} - X_{min}) \qquad (1)$$

Standardized formulae for negative indicators:

$$b = (X_{max} - X)/(X_{max} - X_{min}) \qquad (2)$$

Where $b$ is the standardized evaluation index value; $X$ is the original value of each evaluation

**Table 1. Evaluation index system of WRCC of the UACY.**

| Target Layer | Ruler Layer | Indicator Layer | Unit | Calculation Methods | Indicator Properties | Weights |
|---|---|---|---|---|---|---|
| WRCC | Pressure | population density($X_1$) | person/km$^2$ | total population/total area | negative | 0.0633 |
| | | water consumption of CNY 10,000 of industrial added value($X_2$) | m$^3$ | from the Water Resources Bulletin | negative | 0.0433 |
| | | water consumption per CNY 10,000 GDP($X_3$) | m$^3$ | from the Water Resources Bulletin | negative | 0.0379 |
| | | per capita urban domestic water consumption ($X_4$) | L/d | from the Water Resources Bulletin | negative | 0.0483 |
| | | average annual fertilizer application per unit cultivated land($X_5$) | kg/hm$^2$ | annual fertilizer application/area of arable land | negative | 0.0818 |
| | | agricultural irrigation quota($X_6$) | m$^3$/hm$^2$ | water use for agricultural irrigation/cultivated land area | negative | 0.0530 |
| | | percentage of water used for domestic use by urban and rural residents($X_7$) | % | urban and rural domestic water consumption/ total water consumption | negative | 0.0688 |
| | | per capita urban sewage discharges($X_8$) | m$^3$/Person/ year | annual urban sewage discharge/total population | negative | 0.0456 |
| | State | development and utilization rate of water resources($X_9$) | % | from the Water Resources Bulletin | negative | 0.0588 |
| | | water resources per capita ($X_{10}$) | m$^3$/person | total water resources/total population | positive | 0.0605 |
| | | annual water production modulus ($X_{11}$) | 10$^4$m$^3$/km$^2$ | total water resources/area of the region | positive | 0.0617 |
| | | annual precipitation($X_{12}$) | mm | from the Water Resources Bulletin | positive | 0.0518 |
| | Response | per capita GDP($X_{13}$) | CNY | from statistical data | positive | 0.0770 |
| | | water consumption ratio of the ecological environment($X_{14}$) | % | from the Water Resources Bulletin | positive | 0.1079 |
| | | sewage treatment rate($X_{15}$) | % | from the Water Resources Bulletin | positive | 0.0503 |
| | | vegetation coverage ratio($X_{16}$) | % | from the State of the Ecosystem Bulletin | positive | 0.0900 |

index; and, respectively, $X_{max}$ and $X_{min}$ are the maximum and minimum values in the selected sample series.

*(2) Determination of indicator weights using the entropy weighting method.* Entropy weight method is a method of calculating the weights of evaluation indexes in information theory, which uses the magnitude of the information provided by the entropy value of each evaluation index to calculate the weights of the indexes under objective conditions. When the information entropy of the evaluation indicators is larger, the more balanced the indicator data are, indicating that the indicator's role in influencing the evaluated object is smaller, and the weight it occupies is smaller. On the contrary, the smaller the entropy value, the greater the difference of the indicator data, indicating that the indicator has a greater role in influencing the evaluated object, and the greater the weight occupied by the indicator [40]. The entropy weight method can try to avoid the interference of subjective factors in the calculation process, so that the weight calculation of evaluation indicators is more objective and scientific. The specific calculation steps are as follows.

Step 1: The evaluation indicator values are normalized and the formula is as follows.

$$P_{ij} = \frac{X_{ij}}{\sum_{i=1}^{m} X_{ij}} \tag{3}$$

Where $P_{ij}$ is the proportion of the standardized value of the *j*-th indicator in the *i*-th year, the study selects the relevant data from 2008–2020, $1 \leq m \leq 13$.

Step 2: Calculate the information entropy of the evaluation index, the formula is as follows.

$$E_j = -k\sum_{i=1}^{m}(P_{ij}\ln P_{ij}) \tag{4}$$

Where $E_j$ denotes the information entropy of each indicator, $k = 1/\ln(m)$, and $m$ is the sample region in which the evaluation indicator values are located.

Step 3: Calculate the weight of each indicator with the following formula.

$$W_j = \frac{1 - E_j}{\sum\limits_{j=1}^{n}(1 - E_j)} \tag{5}$$

Where $W_j$ is the weight of each indicator and $n$ is the number of indicators.

**Comprehensive evaluation using set-pair analysis.** Set-pair analysis is an effective systematic analysis method for solving uncertainty problems, which was proposed by Chinese scholar Keqin Zhao in 1989, and its principle is a systematic and mathematical analysis of the certainty and uncertainty of two sets in a set-pair and the interaction between them [41, 42]. The set-pair analysis method can deeply and comprehensively analyze and quantitatively express the information on the degree of conformity between the evaluation sample data and the evaluation level, reflecting the complex system structure of the set-pair of WRCC evaluation samples and evaluation level criteria. The main expressive tool of set-pair analysis is the linking number, which is used to describe the determinism and uncertainty of the system. The set-pair analysis method can accurately reflect the evolution of WRCC and the internal mechanism.

The linkage number is a quantitative indicator that characterizes the degree of comprehensive association between sets in set-pair analysis, and is usually expressed as $\mu$. The formula is as follows.

$$\mu_{A-B} = \frac{S}{N} + \frac{F}{N}i + \frac{P}{N}j = a + b_i + c_j \tag{6}$$

Where $\mu_{A-B}$ is the degree of connectedness, $S$ is the number of sameness in set pairs, $F$ is the number of differences in set pairs, $P$ is the number of opposites in set pairs, $N$ is the total number of features, $a$ denotes the degree of sameness, $b$ denotes the degree of difference, $c$ denotes the degree of opposites, $a + b + c = 1$.

Expanding $b_i$ in Eq (6) further to $b_i = b_1i_1 + b_2i_2 + \ldots$ can obtain the multivariate linkage number, which refines the degree of difference and increases the accuracy of the calculation results when analyzing the uncertainties in the system. The number of terms for the expansion of the degree of discrepancy depends on the number of system evaluation levels, and the number of quintuple linkages divides the evaluation index system into five levels, and the number of terms for the expansion of the degree of discrepancy $b_i$ term is 3, that is:

$$\mu_{A-B} = a + b_1i_1 + b_2i_2 + b_3i_3 + c_j \tag{7}$$

$a$, $b_1$, $b_2$, $b_3$, and $c$ are expressed as the degree of connection between the indicators to be evaluated and the criteria of evaluation levels I, II, III, IV, and V. $a$ is the positive component of the same measure; $b_1$ is the bias-positive component of the difference measure; $b_2$ is the centered component of the difference measure; $b_3$ is the bias-negative component of the difference measure; and $c$ is the negative component of the opposing measure.

When evaluating the WRCC, the basic data of WRCC to be evaluated are taken as set $A$, and the set of criteria for evaluating the WRCC is taken as set $B$, and the set pair $H = (A, B)$ is

**Table 2. Criteria for evaluating the WRCC.**

| Target Layer | Ruler Layer | Indicator Layer | Surplus bearing (I) | Suitable bearing (II) | Near overload (III) | Mild overload (IV) | Serious overload (V) |
|---|---|---|---|---|---|---|---|
| WRCC | Pressure | $X_1$ | 0~100 | 100~200 | 200~300 | 300~400 | >400 |
| | | $X_2$ | 0~30 | 30~60 | 60~90 | 90~120 | >120 |
| | | $X_3$ | 0~40 | 40~50 | 50~60 | 60~70 | >70 |
| | | $X_4$ | 0~120 | 120~150 | 150~180 | 180~220 | 220~350 |
| | | $X_5$ | 0~200 | 200~400 | 400~600 | 600~800 | >800 |
| | | $X_6$ | 0~3000 | 3000~4500 | 4500~6000 | 6000~7500 | >7500 |
| | | $X_7$ | 0~0.8 | 0.8~0.16 | 0.16~0.24 | 0.24~0.32 | >0.32 |
| | | $X_8$ | 0~30 | 30~60 | 60~90 | 90~120 | >120 |
| | State | $X_9$ | 0~10 | 10~20 | 20~30 | 30~40 | >40 |
| | | $X_{10}$ | >3000 | 1700~3000 | 1000~1700 | 500~1000 | <500 |
| | | $X_{11}$ | >60 | 45~60 | 30~45 | 15~30 | <15 |
| | | $X_{12}$ | >1250 | 1100~1250 | 950~1100 | 800~950 | 0~800 |
| | Response | $X_{13}$ | >180000 | 120000~180000 | 30000~120000 | 12000~30000 | 0~12000 |
| | | $X_{14}$ | 5.5~15 | 4~5.5 | 2.5~4 | 1~2.5 | 0~1 |
| | | $X_{15}$ | 95~100 | 90~95 | 80~90 | 80~70 | 0~70 |
| | | $X_{16}$ | 80~100 | 60~80 | 40~60 | 20~40 | 0~20 |

constructed. Based on the reference to "Yunnan Provincial Statistical Yearbook", "Environmental Quality Standard for Surface Water", "Yunnan Provincial Water Resources Planning", WRCC evaluation standard, standards issued by local governments and domestic and international WRCC evaluation, this study divides WRCC into five levels through field investigation and expert consultation: Level I, "surplus bearing", indicating abundant water resources; Level II, "suitable bearing", indicating coordinated use of water resources; Level III, "near overload", indicating water resources constraints; Level IV, "mild overload", indicating water resources shortage; and Level V, "serious overload", indicating severe water resources shortage. The specific grading criteria are shown in Table 2.

The indicators selected for evaluating the WRCC in this study are categorized into positive and negative indicators.

When smaller data for an indicator indicates better data, the degree of association $\eta(X_i, B_k)$ between the sample value $X$ and the $k$-level evaluation criterion for the indicator is calculated as follows.

$$\begin{cases} 1 + 0i_1 + 0i_2 + 0i_3 + 0j; & x_l \leq s_1 \\ \dfrac{s_1 + s_2 - 2x_1}{s_2 - s_1} + \dfrac{2x - 2s_1}{s_2 - s_1}i_1 + 0i_2 + 0i_3 + 0j; & s_1 < x \leq \dfrac{s_1 + s_2}{2} \\ 0 + \dfrac{s_2 + s_3 - 2x}{s_3 - s_1}i_1 + \dfrac{2x - s_1 - s_2}{s_3 - s_1}s_2 + 0i_3 + 0j; & \dfrac{s_1 + s_2}{2} < x \leq \dfrac{s_2 + s_3}{2} \\ 0 + 0i_1 + \dfrac{s_3 + s_4 - 2x}{s_4 - s_2}i_2 + \dfrac{2x - s_2 - s_3}{s_4 - s_2}i_3 + 0j; & \dfrac{s_2 + s_3}{2} < x \leq \dfrac{s_3 + s_4}{2} \\ 0 + 0i_1 + 0i_2 + \dfrac{2s_4 - 2x}{s_4 - s_3}i_3 + \dfrac{2x - s_3 - s_4}{s_4 - s_3}j; & \dfrac{s_3 + s_4}{2} < x \leq s_4 \\ 0 + 0i_1 + 0i_2 + 0i_3 + 1j; & x_l > s_4 \end{cases} \tag{8}$$

When larger data for an indicator indicates better data, the degree of association $\eta(X_i, B_k)$ between the sample value $X$ and the $k$-level evaluation criterion for the indicator is calculated

as follows.

$$
\begin{cases}
1 + 0i_1 + 0i_2 + 0i_3 + 0j; & x_l \geq s_1 \\[2mm]
\dfrac{2x - s_1 - s_2}{s_1 - s_2} + \dfrac{2s_1 - 2x_1}{s_1 - s_2}i_1 + 0i_2 + 0i_3 + 0j; & \dfrac{s_1 + s_2}{2} \leq x < s_1 \\[2mm]
0 + \dfrac{2x - s_3 - s_4}{s_1 - s_3}i_1 + \dfrac{s_1 + s_2 - 2x}{s_1 - s_3}i_2 + 0i_3 + 0j; & \dfrac{s_2 + s_3}{2} \leq x < \dfrac{s_1 + s_2}{2} \\[2mm]
0 + 0i_1 + \dfrac{2x - s_3 - s_4}{s_2 - s_4}i_2 + \dfrac{s_2 + s_3 - 2x}{s_2 - s_4}i_3 + 0j; & \dfrac{s_3 + s_4}{2} \leq x < \dfrac{s_2 + s_3}{2} \\[2mm]
0 + 0i_1 + 0i_2 + \dfrac{2x - 2s_4}{s_3 - s_4}i_3 + \dfrac{s_3 + s_4 - 2x}{s_3 - s_4}j; & s_4 \leq x < \dfrac{s_3 + s_4}{2} \\[2mm]
0 + 0i_1 + 0i_2 + 0i_3 + 1j; & x < s_4
\end{cases}
\tag{9}
$$

Where $S_1$, $S_2$, $S_3$ and $S_4$ are the boundary values of the WRCC for surplus bearing, suitable bearing, near overload, mild overload and serious overload, respectively.

The composite linkage degree of the five-element linkage number can be calculated by the following formula, and the confidence criterion is used to determine the evaluation level of WRCC.

$$
\mu_{A-B} = W\mu_{APBQ}E^T = (w_1, w_2, \ldots w_m)
\begin{bmatrix}
\mu_{11} & \mu_{12} & \cdots & \mu_{15} \\
\mu_{21} & \mu_{22} & \cdots & \mu_{25} \\
\vdots & \vdots & \ddots & \vdots \\
\mu_{m1} & \mu_{m2} & \cdots & \mu_{m5}
\end{bmatrix}
\begin{pmatrix}
1 \\
i_1 \\
i_2 \\
i_3 \\
j
\end{pmatrix}
=
\tag{10}
$$

$$
\sum\nolimits_{p=1}^{m} w_p\mu_{p1} + \sum\nolimits_{p=1}^{m} w_p\mu_{p2}i_1 + \sum\nolimits_{p=1}^{m} w_p\mu_{p3}i_2 + \sum\nolimits_{p=1}^{m} w_p\mu_{p4}i_3 + \sum\nolimits_{p=1}^{m} w_p\mu_{p5}j_3i_3
$$

Let $f_1 = \sum\nolimits_{p=1}^{m} w_p\mu_{p1}, f_2 = \sum\nolimits_{p=1}^{m} w_p\mu_{p2}, f_3 = \sum\nolimits_{p=1}^{m} w_p\mu_{p3}, f_4 = \sum\nolimits_{p=1}^{m} w_p\mu_{p4},$
$f_5 = \sum\nolimits_{p=1}^{m} w_p\mu_{p5}$, Then Eq (10) changes to:

$$
\mu_{A-B} = f_1 + f_2I_1 + f_3I_2 + f_4I_3 + f_5J
\tag{11}
$$

$$
h_k = (f_1 + f_2 + \cdots + f_k) > \lambda, k = 1, 2, 3, 4, 5
\tag{12}
$$

Where $f_1, f_2, f_3, f_4, f_5$ are the linkage component, $f_1$ is the possible degree of affiliation to the level I standard; $f_2$ is the possible degree of affiliation to the level II standard; $f_3$ is the possible degree of affiliation to the level III standard; $f_4$ is the possible degree of affiliation to the level IV standard and $f_5$ is the possible degree of affiliation to the level V standard. $\lambda$ is the degree of confidence, and usually, $0.5 \leq \lambda \leq 0.7$, and for the given $\lambda$, when $h_{k-1} < \lambda$ and $h_k > \lambda$, the evaluation result is level $k$ [43].

**Diagnostic models of barrier degree.** The diagnostic model of handicap degree is a pathological diagnosis of the level or degree of the evaluation target, identifying the main impairment factors [44, 45]. In this study, a barrier degree model was applied to measure the barrier degree of the WRCC indicator layer, which is of great significance in guiding and adjusting the direction of water resources use. There are three main measures of handicap, namely, factor contribution, indicator deviation and handicap. The model is calculated as

follows [46].

$$F_j = W_j \times U_{ij} \tag{13}$$

$$E_j = 1 - K_j \tag{14}$$

$$O_j = F_j \times E_j \Big/ \sum_{j=1}^{m}(F_j \times E_j) \tag{15}$$

$$R_i = \sum_{j=1}^{m} O_{ij} \tag{16}$$

Where $F_j$ denotes the factor contribution; $W_j$ denotes the weight of the $j$-th factor layer; $U_{ij}$ denotes the weight of the $j$-th individual indicator of the $i$-th factor layer; $E_j$ denotes the indicator deviation; $K_j$ denotes the value of the $j$-th indicator after standardization; $O_j$ denotes the degree of obstacle to the improvement of WRCC system of the $j$-th indicator, and m denotes the number of the indicators; $O_{ij}$ denotes the degree of obstacle to the improvement of the WRCC system of the $j$-th indicator of the $i$-th factor layer, and $R_i$ denotes the degree of obstacle to the improvement of the WRCC of the $i$-th factor layer.

## Results and discussion

### Evaluation of water resources carrying capacity

**Evaluation results.** The relevant indicator data of Kunming, Qujing, Yuxi, Chuxiong and Honghe from 2008 to 2020 were brought into Eqs (1) to (5) to obtain the weight values of the evaluation indicators (refer to Table 1). Take Kunming 2020 as an example to introduce the relevant arithmetic process. Firstly, the raw data of 2020 are brought into Eqs (8) and (9) to find out the linkage degree and linkage matrix of each relevant indicator in 2020 (refer to Table 3), and then the weights, linkage matrix and linkage coefficient of each indicator are combined and the linkage value of each indicator is brought into Eq (10) to calculate the linkage degree of each evaluation level in 2020. According to Eq (12), taking $\lambda = 0.5$, the confidence values in 2020 are: $h_1 = f_1 = 0.1515 < \lambda$, $h_2 = f_1 + f_2 = 0.1515 + 0.1151 = 0.2666 < \lambda$, $h_3 = f_1 + f_2 + f_3 = 0.1515 + 0.1151 + 0.3169 = 0.5835 > \lambda$. According to the confidence criterion, it can be concluded that the WRCC grade of Kunming City in 2020 belongs to the third grade, which is on the verge of overloading. Similarly, the WRCC of Kunming City from 2008 to 2020 (refer to Table 4) and the rank of the subsystem carrying capacity in each year (refer to Table 5) are obtained.

**Table 3. Linkage of indicators in Kunming 2020.**

| connectedness | $B_1$ | $B_2$ | $B_3$ | $B_4$ | $B_5$ | connectedness | $B_1$ | $B_2$ | $B_3$ | $B_4$ | $B_5$ |
|---|---|---|---|---|---|---|---|---|---|---|---|
| $\mu_1$ | 0 | 0 | 0 | 0.158 | 0.842 | $\mu_9$ | 0 | 0 | 0.680 | 0.320 | 0 |
| $\mu_2$ | 1 | 0 | 0 | 0 | 0 | $\mu_{10}$ | 0 | 0 | 0 | 0.052 | 0.948 |
| $\mu_3$ | 1 | 0 | 0 | 0 | 0 | $\mu_{11}$ | 0 | 0 | 0 | 0.752 | 0.248 |
| $\mu_4$ | 0 | 0.2 | 0.8 | 0 | 0 | $\mu_{12}$ | 0 | 0 | 0 | 0 | 1 |
| $\mu_5$ | 0 | 0 | 0.394 | 0.606 | 0 | $\mu_{13}$ | 0 | 0.074 | 0.926 | 0 | 0 |
| $\mu_6$ | 0 | 0 | 0.870 | 0.130 | 0 | $\mu_{14}$ | 0.185 | 0.815 | 0 | 0 | 0 |
| $\mu_7$ | 0 | 0 | 0 | 0 | 1 | $\mu_{15}$ | 1 | 0 | 0 | 0 | 0 |
| $\mu_8$ | 0 | 0 | 0.230 | 0.770 | 0 | $\mu_{16}$ | 0 | 0.131 | 0.869 | 0 | 0 |

**Table 4. Linkage degree and $h_k$ value of comprehensive evaluation of WRCC in Kunming, 2008–2020.**

| year | $f_1$ | $f_2$ | $f_3$ | $f_4$ | $f_5$ | $h_1$ | $h_2$ | $h_3$ | $h_4$ | $h_5$ | carrying capacity |
|------|-------|-------|-------|-------|-------|-------|-------|-------|-------|-------|-------------------|
| 2008 | 0.0327 | 0.0845 | 0.3361 | 0.4153 | 0.1315 | 0.0327 | 0.1172 | 0.4532 | 0.8685 | 1 | mild overload |
| 2009 | 0.0000 | 0.1118 | 0.3128 | 0.4193 | 0.1563 | 0.0000 | 0.1118 | 0.4245 | 0.8438 | 1 | mild overload |
| 2010 | 0.0527 | 0.1094 | 0.2678 | 0.4720 | 0.0982 | 0.0527 | 0.1621 | 0.4299 | 0.9019 | 1 | mild overload |
| 2011 | 0.0000 | 0.1317 | 0.1946 | 0.1731 | 0.5007 | 0.0000 | 0.1317 | 0.3263 | 0.4994 | 1 | serious overload |
| 2012 | 0.0006 | 0.1285 | 0.3507 | 0.2828 | 0.2375 | 0.0006 | 0.1291 | 0.4798 | 0.7626 | 1 | mild overload |
| 2013 | 0.0000 | 0.1250 | 0.3766 | 0.3951 | 0.1034 | 0.0000 | 0.1250 | 0.5016 | 0.8967 | 1 | near overload |
| 2014 | 0.0032 | 0.1400 | 0.4619 | 0.3593 | 0.0357 | 0.0032 | 0.1432 | 0.6050 | 0.9644 | 1 | near overload |
| 2015 | 0.0354 | 0.2592 | 0.4208 | 0.2333 | 0.0513 | 0.0354 | 0.2946 | 0.7155 | 0.9487 | 1 | near overload |
| 2016 | 0.1231 | 0.1738 | 0.3317 | 0.2765 | 0.0950 | 0.1231 | 0.2969 | 0.6286 | 0.9051 | 1 | near overload |
| 2017 | 0.1458 | 0.1695 | 0.3900 | 0.2295 | 0.0652 | 0.1458 | 0.3153 | 0.7053 | 0.9348 | 1 | near overload |
| 2018 | 0.2065 | 0.0827 | 0.3994 | 0.3004 | 0.0110 | 0.2065 | 0.2892 | 0.6887 | 0.9891 | 1 | near overload |
| 2019 | 0.2409 | 0.0727 | 0.2593 | 0.3212 | 0.1061 | 0.2409 | 0.3136 | 0.5728 | 0.8940 | 1 | near overload |
| 2020 | 0.1515 | 0.1151 | 0.3169 | 0.1699 | 0.2466 | 0.1515 | 0.2667 | 0.5835 | 0.7535 | 1 | near overload |

According to the above calculation of Kunming's WRCC, the WRCC status of Qujing, Yuxi, Chuxiong and Honghe from 2008 to 2020 (refer to Table 6) and the rank of the subsystem's carrying capacity in each year are obtained in the same way.

**Evaluation and analysis of WRCC.** *1. Comprehensive evaluation analysis.* As can be seen from Table 6, the WRCC of the five cities (states) in the UACY in the study period has a similar trend, which can be divided into two time periods. In the first stage, 2008–2011, the WRCC of Kunming City has changed from "mild overload" to "serious overload"; Qujing and Yuxi have changed from "near overload" to "mild overload"; Honghe has changed from "suitable bearing" to "near overload", and Chuxiong shifted from "suitable bearing" to "near overload", and remained "near overload" throughout, but the confidence value of "near overload" is decreasing, which indirectly also indicates that the WRCC of Chuxiong is also deteriorating. During the period 2008–2011, there was a decreasing trend in the WRCC of all five cities in the UACY. The reason for this is that the consecutive years of drought in Yunnan Province from 2009 to 2011 have reduced the total amount of water resources, the water consumption of CNY 10,000 of industrial added value and water consumption per CNY 10,000 GDP is too high, industrial water consumption and wastage are serious, the efficiency and effectiveness of water use is low, and the WRCC has deteriorated for a short period of time. In the second stage, from 2012 to 2020, the WRCC of the five cities (states) are in a fluctuating upward trend, and the WRCC is steadily developing. The reason for this is that with the implementation of the Yunnan Provincial Water Conservation Regulations in 2013, cities (states) have improved their water use efficiency and effectiveness, and the water consumption of CNY 10,000 of industrial added value and water consumption per CNY 10,000 GDP have been gradually reduced each year, but the effect of reducing the average per hectare of water used in agriculture has been less pronounced, and agricultural water-saving technologies need to be strengthened.

From the results of the integrated assessment of WRCC, the water resources of the UACY are facing greater water supply pressure, and the intensity of chemical fertilizer and pesticide application is high, which triggers agricultural surface source pollution. The WRCC can be improved in the following ways: firstly, water conservation should be increased; Kunming City will have greater pressure on domestic water use in the future, increasing the application of water-saving appliances; other municipalities (states) have greater water consumption in agriculture, and need to strictly manage water resources, optimize the layout of agricultural

**Table 5. Evaluation level and $h_k$ value of WRCC subsystem of Kunming City.**

| year | Ruler Layer | $h_1$ | $h_2$ | $h_3$ | $h_4$ | $h_5$ | carrying capacity |
|---|---|---|---|---|---|---|---|
| 2008 | Pressure | 0.0739 | 0.2650 | 0.5116 | 0.8163 | 1 | near overload |
| | State | 0.0000 | 0.0000 | 0.5468 | 0.9999 | 1 | near overload |
| | Response | 0.0000 | 0.0000 | 0.3068 | 0.8453 | 1 | mild overload |
| 2009 | Pressure | 0.0000 | 0.2528 | 0.5793 | 0.9143 | 1 | near overload |
| | State | 0.0000 | 0.0000 | 0.1548 | 0.5945 | 1 | mild overload |
| | Response | 0.0000 | 0.0000 | 0.4727 | 1.0000 | 1 | mild overload |
| 2010 | Pressure | 0.1192 | 0.2650 | 0.4893 | 0.9143 | 1 | mild overload |
| | State | 0.0000 | 0.0000 | 0.0707 | 0.7408 | 1 | mild overload |
| | Response | 0.0000 | 0.1383 | 0.6061 | 1.0000 | 1 | near overload |
| 2011 | Pressure | 0.0000 | 0.1931 | 0.5381 | 0.8887 | 1 | near overload |
| | State | 0.0000 | 0.0000 | 0.1769 | 0.2527 | 1 | serious overload |
| | Response | 0.0000 | 0.1424 | 0.4528 | 0.8539 | 1 | mild overload |
| 2012 | Pressure | 0.0000 | 0.1781 | 0.6286 | 1.0000 | 1 | near overload |
| | State | 0.0000 | 0.0000 | 0.1668 | 0.2551 | 1 | serious overload |
| | Response | 0.0019 | 0.1548 | 0.5015 | 0.8029 | 1 | near overload |
| 2013 | Pressure | 0.0000 | 0.1711 | 0.6341 | 1.0000 | 1 | near overload |
| | State | 0.0000 | 0.0000 | 0.1592 | 0.5557 | 1 | mild overload |
| | Response | 0.0000 | 0.1517 | 0.5663 | 1.0000 | 1 | near overload |
| 2014 | Pressure | 0.0073 | 0.2457 | 0.5535 | 0.9196 | 1 | near overload |
| | State | 0.0000 | 0.0000 | 0.3073 | 0.9991 | 1 | mild overload |
| | Response | 0.0000 | 0.1063 | 0.8881 | 1.0000 | 1 | near overload |
| 2015 | Pressure | 0.0802 | 0.2804 | 0.5905 | 0.8838 | 1 | near overload |
| | State | 0.0000 | 0.0909 | 0.6488 | 0.9999 | 1 | near overload |
| | Response | 0.0000 | 0.4597 | 0.9328 | 1.0000 | 1 | near overload |
| 2016 | Pressure | 0.0343 | 0.3227 | 0.5619 | 0.8517 | 1 | near overload |
| | State | 0.0000 | 0.0000 | 0.3027 | 0.8736 | 1 | mild overload |
| | Response | 0.3319 | 0.4743 | 0.9525 | 1.0000 | 1 | near overload |
| 2017 | Pressure | 0.0856 | 0.3323 | 0.5425 | 0.9109 | 1 | near overload |
| | State | 0.0000 | 0.0472 | 0.6379 | 0.8887 | 1 | near overload |
| | Response | 0.3319 | 0.4840 | 0.9746 | 1.0000 | 1 | near overload |
| 2018 | Pressure | 0.1824 | 0.2930 | 0.5329 | 0.9751 | 1 | near overload |
| | State | 0.0000 | 0.0000 | 0.5576 | 0.9999 | 1 | near overload |
| | Response | 0.3870 | 0.4911 | 0.9941 | 1.0000 | 1 | near overload |
| 2019 | Pressure | 0.1882 | 0.2930 | 0.4735 | 0.8967 | 1 | mild overload |
| | State | 0.0000 | 0.0000 | 0.1642 | 0.7402 | 1 | mild overload |
| | Response | 0.4849 | 0.5658 | 1.0001 | 1.0000 | 1 | suitable bearing |
| 2020 | Pressure | 0.1837 | 0.2055 | 0.4938 | 0.7237 | 1 | mild overload |
| | State | 0.0000 | 0.0000 | 0.1718 | 0.4654 | 1 | mild overload |
| | Response | 0.2162 | 0.5406 | 1.0001 | 1.0000 | 1 | suitable bearing |

production, change the way agriculture uses water, improve agricultural water-saving mechanisms, and focus on strengthening comprehensive measures to save water in agriculture. Secondly, it is necessary to rationally allocate industrial water use. For Kunming, which has a relatively small amount of water resources, it is necessary to actively develop low-water-consuming industries and transfer high-water-consuming industries to areas with richer water resources. Thirdly, we should improve the laws and regulations on water resources management, govern water according to the law, implement the strictest water resources management

**Table 6. Evaluation level of WRCC of the UACY from 2008 to 2020.**

| year | Kunming | Qujing | Yuxi | Chuxiong | Honghe |
|------|---------|--------|------|----------|--------|
| 2008 | mild overload | near overload | near overload | near overload | suitable bearing |
| 2009 | mild overload | near overload | near overload | near overload | near overload |
| 2010 | mild overload | near overload | near overload | near overload | near overload |
| 2011 | serious overload | mild overload | mild overload | near overload | near overload |
| 2012 | mild overload | near overload | near overload | near overload | near overload |
| 2013 | mild overload | near overload | near overload | near overload | suitable bearing |
| 2014 | near overload | near overload | near overload | near overload | near overload |
| 2015 | near overload | near overload | near overload | near overload | near overload |
| 2016 | near overload | near overload | near overload | near overload | suitable bearing |
| 2017 | near overload | near overload | near overload | near overload | suitable bearing |
| 2018 | near overload | near overload | suitable bearing | near overload | suitable bearing |
| 2019 | near overload | near overload | near overload | near overload | near overload |
| 2020 | near overload | near overload | suitable bearing | near overload | suitable bearing |

system, and promote water resources management according to local conditions. Fourthly, the development of eco-agriculture should be vigorously pursued; agricultural production should be based on the application of farmyard manure, supplemented by the application of chemical fertilizers; the prevention and control of agricultural pests and diseases should be based on biological control, with a minimum of pharmaceutical control and the prevention and control of agricultural surface pollution to the maximum extent possible, so as to take the road of ecological and environmentally friendly agricultural development.

The areas with the highest and lowest WRCC in the UACY are dominated by their natural conditions, while the areas with the middle WRCC are dominated by their industrial structure and water resources development and utilization rate. Enhancement of water resources development and utilization, construction of water conservancy projects and rational allocation of water resources can effectively improve the WRCC in the UACY, and the most stringent water resources management system has a long way to go.

*2. Analysis of each evaluation subsystem.* Based on the PSR model, the WRCC of the UACY is mapped into a sub-system carrying capacity as shown in Fig 2, and at the same time, Kunming City, as an example, is identified as a sub-system carrying capacity, and a single-indicator WRCC of Kunming City is mapped out as shown in Fig 3.

*Pressure subsystem*: The carrying capacity of the pressure subsystems in Kunming and Qujing has been decreasing, because with the acceleration of urbanization, the population density and the proportion of water used by urban and rural residents have increased. The improvement in the standard of living has led to an increase in the per capita amount of wastewater discharged. The Yuxi pressure subsystem has evolved from "near overload" to "suitable bearing", as shown in Annex S1 Data, Yuxi's population density, water consumption of CNY 10,000 of industrial added value, and water consumption per CNY 10,000 GDP were lower than Kunming and Qujing, and were in a decreasing trend, which was the main reason for the change in the Yuxi pressure system's carrying capacity to become better. During the study period, the carrying capacity of Chuxiong and Honghe pressure subsystems was basically unchanged, and always maintained "suitable bearing", although the proportion of water used by urban and rural residents and the per capita sewage discharge in the city increased, all other indicators show a decreasing trend, which is the main reason why the pressure systems of the two cities remain "suitable bearing". From the evaluation chart of single indicators in Kunming, it can be seen that in 2008, the single indicators of serious overshoot are water

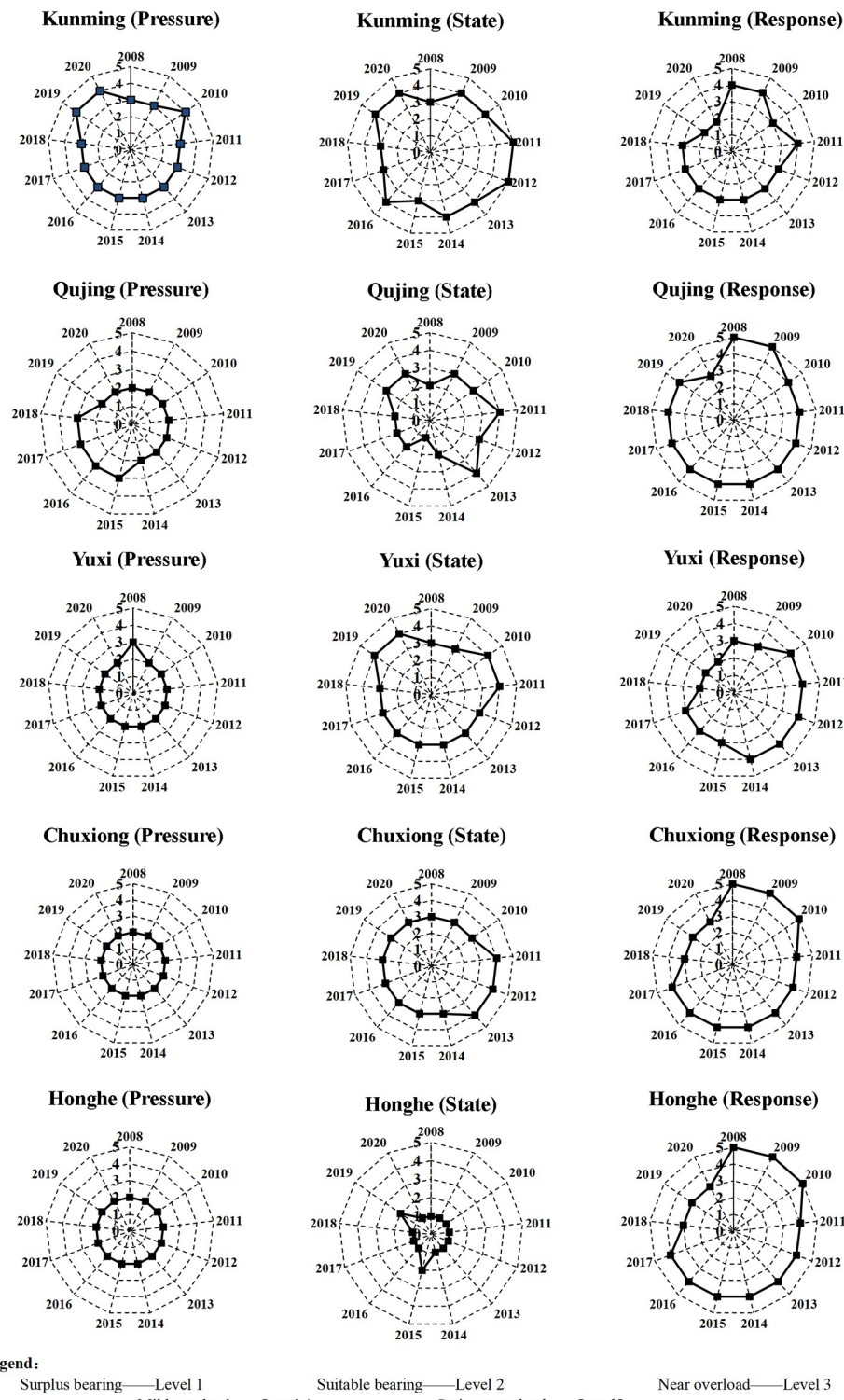

**Fig 2. Evaluation level of WRCC subsystem of the UACY, 2008–2020.**

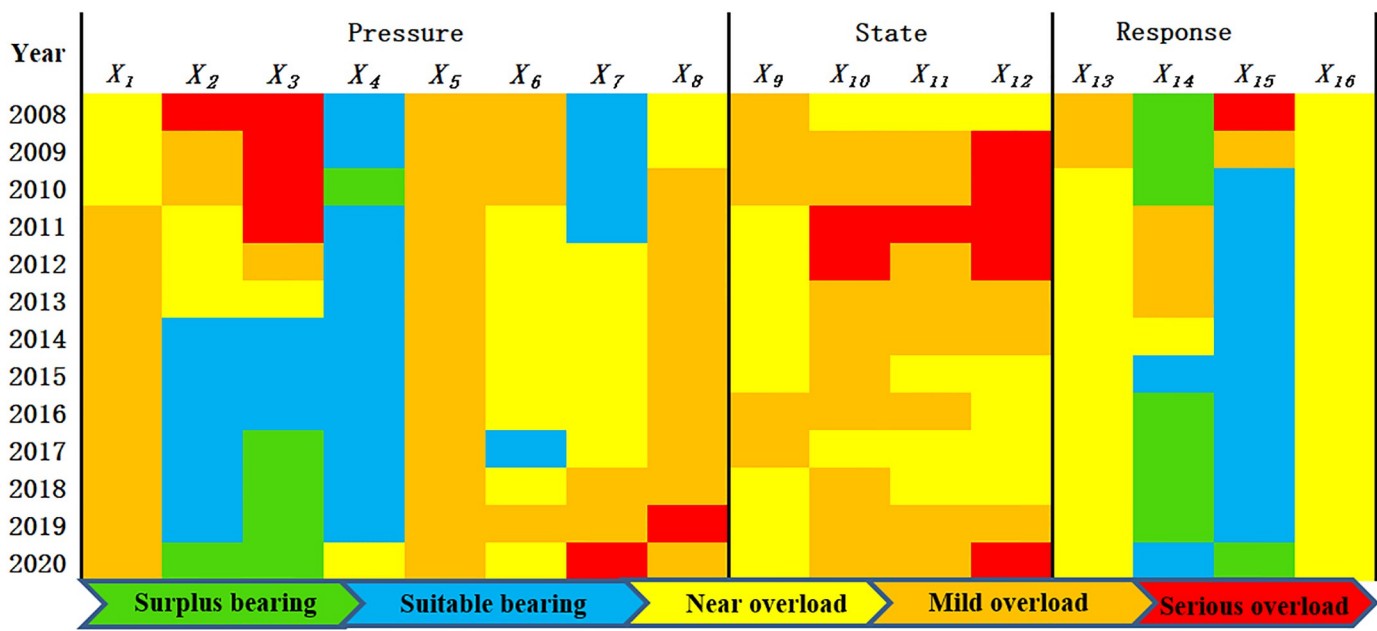

**Fig 3. Evaluation of single indicators for Kunming, 2008–2020.**

consumption of CNY 10,000 of industrial added value and water consumption per CNY 10,000 GDP, and with the improvement of water use efficiency, water consumption of CNY 10,000 of industrial added value and water consumption per CNY 10,000 GDP gradually decrease, and by 2020, the two single indicators have evolved into "surplus bearing". The single indicator of population density in Kunming has evolved from "near overload" to "mild overload", as the population has been growing gradually every year; the single indicator of fertilizer application intensity has been in "mild overload", which in turn has caused agricultural surface source pollution. In the process of urbanization, there has been a massive increase in population, and the indicator of the proportion of water used by urban and rural residents for domestic purposes has evolved from "suitable bearing" in 2008 to "serious overload" in 2020. Per capita wastewater discharges in the Town evolve from "near overload" to "mild overload", peaking in 2019 as "serious overload". Therefore, the main reason for the deteriorating pressure carrying capacity of Kunming is the contradiction between population increase and water resources, which can be improved in terms of controlling population growth and improving the efficiency of resource utilization.

*State subsystem*: The state subsystem of WRCC in Kunming is in "mild overload" and "serious overload". Kunming is the most water-poor city in the UACY, and has been deteriorating to "serious overload" from 2008 to 2012, with a positive trend after 2012. The state subsystem of WRCC in Qujing gradually evolved from "suitable bearing" to "mild overload" from 2008 to 2013, and gradually improved after 2014. The state subsystem of WRCC in Yuxi and Chuxiong fluctuate up and down on the brink of "near overload". Honghe Prefecture has large annual rainfall, abundant water resources and relatively rich water reserves, and its WRCC state subsystem is "Surplus bearing". From the single-indicator state evaluation map of Kunming City, the single-indicator state fluctuates up and down in "mild overload", and the annual precipitation of Kunming City in 2020 is "serious overload", and the state subsystem becomes the main subsystem affecting the WRCC of Kunming. The construction of the

Central Yunnan Urban Agglomeration Diversion Project will alleviate the WRCC status sub-system in Kunming in the future.

*Response subsystem*: From 2008 to 2020, the response subsystem of WRCC in UACY all develops to a good trend. Kunming, as the most economically developed city in the UACY, invests more expenses in water resources management, has a higher sewage treatment rate, and responds in a more-timely manner, which makes Kunming's response subsystem carrying capacity better than the four cities of Qujing, Yuxi, Chuxiong and Honghe. Besides, Qujing and Yuxi's response subsystem carrying capacity is better than Chuxiong and Honghe. In 2020, the response subsystems of Kunming and Yuxi are "suitable bearing", and Qujing, Chuxiong and Honghe are "near overload", and water resource-related governance funds need to be invested more. From the evaluation of single-indicator status in Kunming City, the sewage treatment rate improved greatly during the study period, shifting from "serious overload" to "Surplus bearing"; the forest cover remained consistently "near overload".

## Diagnosis and analysis of water carrying capacity barriers

According to Eqs (13)–(16), the obstacle degree of each indicator and each subsystem of WRCC of each city (state) of the UACY from 2008 to 2020 was calculated. Based on the ranking of obstacle degree, the main obstacle factors affecting the WRCC are identified. Due to the large number of evaluation indicator layers and the limitation of space, only the main obstacle factors with the top 5 obstacle degrees in the four years of 2008, 2013, 2018 and 2020 of the UACY are listed here, as shown in Table 7. The degree of obstacles for each subsystem is shown in Table 8.

Table 7 shows that in 2008, the main obstacles to the WRCC of each region of the UACY were the indicators of water consumption of CNY 10,000 of industrial added value, water consumption per CNY 10,000 GDP, sewage treatment rate, agricultural irrigation quota and other indicators. At that time, the government made great efforts to develop the economy, and the industrial water consumption was on the high side, and the sewage treatment rate was on the low side, and in 2008, the water consumption per CNY 10,000 GDP of Kunming, Qujing, Yuxi, Chuxiong, and Honghe were respectively 135 $m^3$, 179 $m^3$, 166 $m^3$, 362 $m^3$, 322 $m^3$, and the single indicators are all in "serious overload". With the implementation of the strictest water resources management system and the optimization of industrial structure, the water resources utilization rate has been gradually improved, and the water consumption of CNY 10,000 of industrial added value and water consumption per CNY 10,000 GDP have been decreasing year by year, and the main obstacles to WRCC have been changed. In 2013, the situation of obstacle factors in the UACY was different, and the main obstacle factors were the water consumption ratio of the ecological environment, vegetation coverage ratio, average annual fertilizer application per unit cultivated land. The rapid development of agriculture in the UACY has led to the average annual fertilizer application per unit cultivated land becoming the main obstacle factor.

In 2018, the main obstacle factors in Kunming were the average annual fertilizer application per unit cultivated land, per capita urban sewage discharges, and the percentage of water used for domestic use by urban and rural residents; the main obstacle factors to the WRCC of Qujing, Yuxi, Chuxiong, and Honghe evolved into the average annual fertilizer application per unit cultivated land, development and utilization rate of water resources, and the water consumption ratio of the ecological environment, indirectly indicating that the urbanization process in Kunming has led to a large increase in the urban population, and that the water used by urban and rural residents has taken up a major proportion and become the main obstacle factor to the WRCC of Kunming. In 2020, the proportion of water used by urban and rural

**Table 7. List of major obstacle factors in 2008, 2013, 2018 and 2020.**

| Year | | 2008 | | 2013 | | 2018 | | 2020 | |
|---|---|---|---|---|---|---|---|---|---|
| Kunming | Barrier factor 1 | $X_2$ | 0.1538 | $X_{14}$ | 0.1639 | $X_5$ | 0.1425 | $X_7$ | 0.1692 |
| | Barrier factor2 | $X_{13}$ | 0.1479 | $X_8$ | 0.1206 | $X_7$ | 0.1394 | $X_1$ | 0.1558 |
| | Barrier factor3 | $X_3$ | 0.1325 | $X_5$ | 0.1078 | $X_8$ | 0.1279 | $X_{10}$ | 0.1182 |
| | Barrier factor4 | $X_6$ | 0.1018 | $X_{13}$ | 0.0988 | $X_{10}$ | 0.1190 | $X_5$ | 0.1024 |
| | Barrier factor5 | $X_{15}$ | 0.0968 | $X_{12}$ | 0.0962 | $X_{16}$ | 0.0756 | $X_8$ | 0.0956 |
| Qujing | Barrier factor1 | $X_{16}$ | 0.1828 | $X_5$ | 0.1285 | $X_{14}$ | 0.1876 | $X_7$ | 0.1815 |
| | Barrier factor2 | $X_2$ | 0.1572 | $X_7$ | 0.1219 | $X_1$ | 0.1045 | $X_9$ | 0.1737 |
| | Barrier factor3 | $X_6$ | 0.1076 | $X_{16}$ | 0.1155 | $X_5$ | 0.1010 | $X_6$ | 0.1167 |
| | Barrier factor4 | $X_{15}$ | 0.1028 | $X_{11}$ | 0.0978 | $X_7$ | 0.0988 | $X_{11}$ | 0.1012 |
| | Barrier factor5 | $X_8$ | 0.0932 | $X_{10}$ | 0.0970 | $X_9$ | 0.0964 | $X_4$ | 0.1008 |
| Yuxi | Barrier factor1 | $X_{16}$ | 0.1826 | $X_{14}$ | 0.1703 | $X_1$ | 0.1491 | $X_7$ | 0.2203 |
| | Barrier factor2 | $X_3$ | 0.1562 | $X_5$ | 0.1133 | $X_5$ | 0.1435 | $X_{11}$ | 0.1584 |
| | Barrier factor3 | $X_2$ | 0.1194 | $X_{16}$ | 0.0993 | $X_6$ | 0.1261 | $X_{10}$ | 0.1459 |
| | Barrier factor4 | $X_{15}$ | 0.1022 | $X_{13}$ | 0.0836 | $X_7$ | 0.1025 | $X_8$ | 0.1294 |
| | Barrier factor5 | $X_8$ | 0.0926 | $X_4$ | 0.0690 | $X_9$ | 0.0997 | $X_6$ | 0.1007 |
| Chuxiong | Barrier factor1 | $X_2$ | 0.1524 | $X_{14}$ | 0.1590 | $X_{14}$ | 0.1843 | $X_7$ | 0.1842 |
| | Barrier factor2 | $X_{16}$ | 0.1431 | $X_{16}$ | 0.1278 | $X_5$ | 0.1605 | $X_9$ | 0.1499 |
| | Barrier factor3 | $X_3$ | 0.1224 | $X_7$ | 0.1212 | $X_7$ | 0.1137 | $X_5$ | 0.1442 |
| | Barrier factor4 | $X_1$ | 0.0994 | $X_{13}$ | 0.0999 | $X_9$ | 0.1021 | $X_8$ | 0.1321 |
| | Barrier factor5 | $X_9$ | 0.0936 | $X_{10}$ | 0.0923 | $X_{16}$ | 0.0779 | $X_{11}$ | 0.1227 |
| Honghe | Barrier factor1 | $X_2$ | 0.1857 | $X_{14}$ | 0.2029 | $X_5$ | 0.1749 | $X_7$ | 0.1621 |
| | Barrier factor2 | $X_3$ | 0.1695 | $X_{16}$ | 0.1416 | $X_7$ | 0.1489 | $X_5$ | 0.1378 |
| | Barrier factor3 | $X_{13}$ | 0.1449 | $X_{13}$ | 0.1079 | $X_1$ | 0.1480 | $X_8$ | 0.1237 |
| | Barrier factor4 | $X_{15}$ | 0.0948 | $X_5$ | 0.1047 | $X_4$ | 0.1058 | $X_{11}$ | 0.1189 |
| | Barrier factor5 | $X_6$ | 0.0890 | $X_{11}$ | 0.0656 | $X_{13}$ | 0.0948 | $X_4$ | 0.1164 |

residents in Kunming ranked first in the barrier factor, and the proportion of water used by residents exceeded the proportion of water used by industry. In the future, it is recommended to speed up the implementation of the priority strategy of water conservation, establish a water market system with market regulation and user participation, and implement a ladder price for residential water use, so as to rapidly promote the construction of a water-saving society.

As can be seen from Table 8, from the results of the obstacle degree of each subsystem, the impact of the pressure subsystem on the WRCC shows an overall upward trend, the increase in population density and per capita water consumption, resulting in increased pressure on water use and sewage discharge pressure. The impact of the response subsystem on the WRCC shows an overall downward trend, indicating that the optimization of industrial structure has been effective. The effect of the state subsystem on the WRCC fluctuates up and down and is closely related to regional water resource characteristics.

## Conclusions

This study takes the UACY as the case study, constructs the WRCC evaluation index system based on the PSR model, uses the set-pair analysis method and the obstacle degree diagnostic model to evaluate the WRCC of each state from 2008 to 2020 and calculates the main obstacle factors, and the method can be better applied to this study. The study reached the following conclusions.

**Table 8. Degree of obstacles in each subsystem in 2008, 2013, 2018 and 2020.**

| City (state) | Year | Pressure | State | Response |
|---|---|---|---|---|
| Kunming | 2008 | 0.4451 | 0.1719 | 0.3830 |
| | 2013 | 0.4349 | 0.2912 | 0.2739 |
| | 2018 | 0.6051 | 0.2157 | 0.1792 |
| | 2020 | 0.6655 | 0.1894 | 0.1451 |
| Qujing | 2008 | 0.3643 | 0.1926 | 0.4431 |
| | 2013 | 0.4491 | 0.2757 | 0.2752 |
| | 2018 | 0.4590 | 0.1923 | 0.3487 |
| | 2020 | 0.5320 | 0.2020 | 0.2660 |
| Yuxi | 2008 | 0.4005 | 0.1586 | 0.4410 |
| | 2013 | 0.4311 | 0.1717 | 0.3973 |
| | 2018 | 0.6488 | 0.1251 | 0.2262 |
| | 2020 | 0.5378 | 0.2557 | 0.2064 |
| Chuxiong | 2008 | 0.3871 | 0.1156 | 0.4973 |
| | 2013 | 0.2978 | 0.2809 | 0.4213 |
| | 2018 | 0.4122 | 0.2555 | 0.3323 |
| | 2020 | 0.5407 | 0.2493 | 0.2100 |
| Honghe | 2008 | 0.3397 | 0.0654 | 0.5950 |
| | 2013 | 0.3384 | 0.1696 | 0.4919 |
| | 2018 | 0.7092 | 0.0747 | 0.2161 |
| | 2020 | 0.6138 | 0.2104 | 0.1758 |

1. The WRCC of the UACY presents a stage-by-stage evolution in the time dimension, with a decreasing trend of carrying capacity from 2008 to 2012, during which the state subsystem has the greatest impact; from 2013 to 2020, the overall WRCC shows a fluctuating upward growth trend. In the spatial dimension, Kunming's water overload situation is more serious than that of Qujing, Yuxi, Chuxiong and Honghe. The water use structure of Kunming has developed from "industrial and agricultural water use is dominant" to "the proportion of industrial water use has decreased, the proportion of agricultural water use has stabilized, and the proportion of domestic water use has increased". The water use structure of Qujing, Yuxi, Chuxiong and Honghe has not changed much. With the gradual implementation of water conservation measures, the overall utilization efficiency of water resources has gradually improved, and the water consumption of CNY 10,000 of industrial added value and water consumption per CNY 10,000 GDP have shown a continuous downward trend, and the utilization efficiency of industrial water has improved significantly; the change in the efficiency of agricultural water use is not so obvious, and the future of agricultural water conservation measures need to be strengthened.

2. From the WRCC evaluation sub-system, the carrying capacity of the pressure sub-system in Kunming and Qujing shows gradual deterioration. The affect of the state subsystem on the WRCC is closely related to the characteristics of regional water resources. The response subsystems are all moving toward a good trend. Kunming, as the most economically developed city in the UACY, has invested more money in water resource management, has a higher wastewater treatment rate, and responds in a more-timely manner, which makes Kunming's response subsystem carrying capacity better than the four cities of Qujing, Yuxi, Chuxiong and Honghe. Improving the utilization rate of water resources development, constructing water conservancy projects, and rational allocation of water resources can effectively improve the WRCC in the UACY.

3. The WRCC varies from region to region and is constrained by different factors. In 2008, the main obstacles to the WRCC in each region were the water consumption of CNY 10,000 of industrial added value, water consumption per CNY 10,000 GDP, the sewage treatment rate, and the average annual fertilizer application per unit cultivated land. In 2018, the main obstacle factors in Kunming were the average annual fertilizer application per unit cultivated land, urban per capita sewage discharge, percentage of water used for domestic use by urban and rural residents; the main obstacle factors to the WRCC of Qujing, Yuxi, Chuxiong and Honghe evolved to be the average annual fertilizer application per unit cultivated land, the development and utilization rate of water resources, and the water consumption ratio of the ecological environment.

4. From the results of the subsystem barrier degree, it can be concluded that the impact of the pressure subsystem on the WRCC has an overall increasing trend. The response subsystem showed an overall decreasing trend in impacts on the WRCC. The effect of the state subsystem on the WRCC fluctuates up and down and is closely related to the characteristics of regional water resources. The systems theory should be used to promote the virtuous development of water security.

This study takes economic, social, resource, environmental and ecological factors into consideration, analyses the temporal and spatial patterns of change in the WRCC of the UACY, and in order to further improve the WRCC, the government should strengthen infrastructure construction and optimize the industrial layout, improve the utilization rate of water resources through multiple channels, carry out water pollution prevention and control, promote ecological environmental protection, and change the mode of economic development, so as to guarantee the sustainable development of the economy and society. Due to the limitation of raw data, this study is conducted from the city (state) level, and the value of the study would be more prominent if the sample data of 49 counties (districts) in the UACY were used as the case of the study. It would be more conducive to the sustainable development of urban agglomerations if big data tracking and monitoring, simulation and prediction can be used for the study of WRCC. Using remote sensing image data to obtain more accurate data, such as based on ArcGIS software to obtain multi-period land use data, DEM data, POI point data, nighttime lighting data, population density, per capita GDP, and other economic, social and environmental indicators, to construct raster data with an accuracy of 30m to analyze and predict the WRCC of UACY. This will further improve the accuracy of the data acquisition and evaluation model, and provide micro-level suggestions for the improvement of water resources carrying capacity. Of course, this is also the direction of future research.

## Supporting information

**S1 Data.**
(XLSX)

## Acknowledgments

Thanks to all those who helped in the manuscript writing process. We thank the reviewers for their useful comments and suggestions.

## Author Contributions

**Conceptualization:** Jing Zhang.

**Data curation:** Wenchuang Guan, Jing Wang.

**Formal analysis:** Jing Zhang, Bulin Zhang.

**Funding acquisition:** Jing Wang.

**Investigation:** Guangping Wu.

**Methodology:** Wenchuang Guan, Biyu Rao.

**Project administration:** Jing Wang.

**Resources:** Biyu Rao.

**Software:** Jing Zhang, Wenchuang Guan.

**Supervision:** Guangping Wu.

**Validation:** Wenchuang Guan.

**Visualization:** Jing Zhang, Bulin Zhang.

**Writing – original draft:** Jing Zhang, Jing Wang.

**Writing – review & editing:** Jing Zhang, Wenchuang Guan, Jing Wang.

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
