## [Decision Letter · Decision Letter 0]

9 Apr 2024

PONE-D-24-09140Study on water resources carrying capacity based on Pressure-State-Response modeling：An empirical study of the urban agglomeration in Central Yunnan, ChinaPLOS ONE

Dear Dr. Wang,

Thank you for submitting your manuscript to PLOS ONE. After careful consideration, we feel that it has merit but does not fully meet PLOS ONE’s publication criteria as it currently stands. Therefore, we invite you to submit a revised version of the manuscript that addresses the points raised during the review process.

We look forward to receiving your revised manuscript.

Kind regards,

Sivasankar Koppala

Academic Editor

PLOS ONE

Journal Requirements:

"Thanks to all those who helped in the manuscript writing process. We thank the reviewers for their useful comments and suggestions. Thanks to the National Natural Science Foundation of China for funding this project."

3.Thank you for stating the following financial disclosure:

4. We note that Figure 1 in your submission contain map images which may be copyrighted. All PLOS content is published under the Creative Commons Attribution License (CC BY 4.0), which means that the manuscript, images, and Supporting Information files will be freely available online, and any third party is permitted to access, download, copy, distribute, and use these materials in any way, even commercially, with proper attribution. For these reasons, we cannot publish previously copyrighted maps or satellite images created using proprietary data, such as Google software (Google Maps, Street View, and Earth). For more information, see our copyright guidelines: http://journals.plos.org/plosone/s/licenses-and-copyright.

   a. You may seek permission from the original copyright holder of Figure(s) 1to publish the content specifically under the CC BY 4.0 license. 

Please upload the completed Content Permission Form or other proof of granted permissions as an "Other" file with your submission

6. Please include captions for your Supporting Information files at the end of your manuscript, and update any in-text citations to match accordingly. Please see our Supporting Information guidelines for more information: http://journals.plos.org/plosone/s/supporting-information

Reviewers' comments:

Reviewer's Responses to Questions

**Comments to the Author**

1. Is the manuscript technically sound, and do the data support the conclusions?

Reviewer #1: Partly

Reviewer #2: Yes

2. Has the statistical analysis been performed appropriately and rigorously? 

Reviewer #1: I Don't Know

Reviewer #2: Yes

3. Have the authors made all data underlying the findings in their manuscript fully available?

Reviewer #1: Yes

Reviewer #2: Yes

4. Is the manuscript presented in an intelligible fashion and written in standard English?

Reviewer #1: No

Reviewer #2: Yes

5. Review Comments to the Author

Reviewer #1: Dear Authors,

The MS titled "Study on water resources carrying capacity based on Pressure-State-Response modeling：An empirical study of the urban agglomeration in Central Yunnan, China" tries to assess the WRCC of an urban system in China. I can't justify the scientific merits of the current version due to multiple comments explained below. Therefore, I try to mention them to improve the quality of the work with a major review decision.

I see five major comments followed by examples of minor corrections.

(1)

WRCC based on the origins of "carrying capacity" concept can have a rigorous physical meaning (e.g., maximum population can live in an area sustainably). Therefore, to address WRCC in this way multiple methods and approaches have been developed.

The other way of looking at WRCC and CC is using system of indicators. Although acceptable, this way of estimating WRCC is used mainly for the data scarce situations.

My main comment is why the authors, while having sophisticated input datasets, choose the system of indicators?

For a physical approach of looking at WRCC, please take a look at below references (as a few examples but not limited to):

https://doi.org/10.1016/j.scitotenv.2022.153038

https://doi.org/10.1016/j.landusepol.2010.01.006

https://eprints.qut.edu.au/67485/

https://doi.org/10.1016/j.jclepro.2017.02.110

In my opinion, the manuscript didn't explain enough why the current specific method was chosen, and the text limits the readers just to a group of publications, with high similarity to the current manuscript.

Also, at the end of MS for discussing the results, it doesn't connect the readers to the physics behind the limits on WRCC.

On the other hand, if the system of indicators is still the chosen method, please take a look at alternatives in the following book and explain your choice.

Applied Panarchy: Applications and Diffusion across Disciplines (Page 59 to 68).

(2)

There is a high level of math and system of equations for the methods and analysis. However, I think the main objective is lost. The main goal is to assess the capacity of the land based on its resources. For a successful use of equations, I think the steps need more clear interpretations. For example, take a look at these two examples:

https://doi.org/10.1126/science.7618100

https://doi.org/10.1016/j.heliyon.2023.e15079

(3)

The PSR, is one of the analytical framework to assess a complex SES systems. In a few sentences explain the existence of others. Also, the more comprehensive way is to use DPSIR framework. Please justify why the MS narrow it down to PSR framework. The following reference can help

https://www.jstor.org/stable/26269404

(4)

Please use the standard and acceptable terminology for the concepts. For example "overload" in Line 287 should be replaced with "overshoot". Or, "the state of carrying" in Line 284 is not a correct term. Please rephrase. Similar examples exist in the MS (Lines 311, 325, 333 and multiple other places).

(5)

In Lines 258-260 the MS says "water-saving technologies in agriculture, adopting sprinkler irrigation, tube irrigation and other irrigation methods."

This argument is not true. The irrigation technologies can save water at the farming scale (small scale). However, at the watershed scale and for the large scale planning, the promotion of irrigation technologies increase the net water use duo to rebound effect. The irrigation technologies improve the water productivity. Please take a look at the FAO report on this. Also, for the scentific basis of this argument, take a look at following references:

https://doi.org/10.1016/j.ecolecon.2005.03.020

https://doi.org/10.1016/j.jclepro.2009.08.001

Besides the above major comments, there are comments on the writing (major and minor).

- The English writing of the MS can be improved significantly.

- Avoid using "etc." in the MS.

- Line 352, do you mean "case study" by the word "research object"?

- Line 371: what do you mean by "which makes Kunming better than the other four cities and states"?

- Line 358: by the word "development", do you mean "growth"?

- Line 384: the word "dimention" is unnecessary.

- Lines 391-393: Why it was not possible to do it in the current study? The use of big data for WRCC is applied before, and the spatio-temporal evolution of the system has been performed internationally and in China. Please mention them in the literature review, and for this paragraph be more specific about the future directions.

Kind regards

Reviewer #2: Water resources carrying capacity is an important indicator to measure the sustainable development of cities and regional water security.The research constructed a water resources carrying capacity evaluation index system based on the PSR model, and evaluated the water resources carrying capacity by using the set-pair analysis method and the obstacle diagnostic model,which has important reference significance. The following suggestions are put forward, which can be changed according to the author and the understanding of the article:

1.The introduction should present the research question through the research synthesis, which leads to a description of the research and research innovations. The research question is not clearly stated in the introduction of the thesis, and the presentation of the research question and the description of the innovation should be improved.

2.It is recommended that the basis for the selection of the indicators for the evaluation of the carrying capacity of water resources in Table 1 be explained.

3.In the discussion section, targeted policy recommendations are made based on the specific situation of each city (state) in the Central Yunnan Urban Agglomeration, the reasons for the differences in water resources carrying capacity, and the main obstacle factors, in order to improve the practical guidance of the article.

4.In lines 296-297：According to the results of Fig. 2 and Table 6, the subsystem of water resources carrying capacity state in Kunming should be between the mild overload and severe overload state, and the description is problematic.

5.The language of this manuscript needs to be improved.

6. PLOS authors have the option to publish the peer review history of their article (what does this mean?). If published, this will include your full peer review and any attached files.

Reviewer #1: No

Reviewer #2: No

---

## [Author Response · Author response to Decision Letter 0]

12 Jun 2024

Response to "Journal Requirements"：

Question 1. 

Response 1.

We have checked the formatting to confirm compliance with PLOS ONE.

Question 2. 

Thank you for stating the following in the Acknowledgments Section of your manuscript:

"Thanks to all those who helped in the manuscript writing process. We thank the reviewers for their useful comments and suggestions. Thanks to the National Natural Science Foundation of China for funding this project."

Response 2.

We have removed any funding-related text from the manuscript.

Question 3. 

Thank you for stating the following financial disclosure:

Response 3.

This study was funded by the National Natural Science Foundation of China (No. 54322066027). 

Question 4.

We note that Figure 1 in your submission contain map images which may be copyrighted. All PLOS content is published under the Creative Commons Attribution License (CC BY 4.0), which means that the manuscript, images, and Supporting Information files will be freely available online, and any third party is permitted to access, download, copy, distribute, and use these materials in any way, even commercially, with proper attribution. For these reasons, we cannot publish previously copyrighted maps or satellite images created using proprietary data, such as Google software (Google Maps, Street View, and Earth). For more information, see our copyright guidelines: http://journals.plos.org/plosone/s/licenses-and-copyright.

 a. You may seek permission from the original copyright holder of Figure(s) 1to publish the content specifically under the CC BY 4.0 license. 

Please upload the completed Content Permission Form or other proof of granted permissions as an "Other" file with your submission

The Gateway to Astronaut Photography of Earth (public domain): http://eol.jsc.nasa.gov/sseop/ clickmap/

Maps at the CIA (public domain): https://www.cia.gov/library/publications/the-world-factbook/ index.html and https://www.cia.gov/library/publications/cia-maps-publications/index.html

USGS EROS (Earth Resources Observatory and Science (EROS) Center) (public domain): http://eros.usgs.gov/

Response 4.

The map sources and written permission from the copyright holder are as follows.

According to the requirements of the official website of data download, we have adjusted the data citations in the manuscript and added data citations to the references as references [31]. The related information is as follows:

(1) Direct link to the source of the base map: https://www.resdc.cn/DOI/DOI.aspx?DOIID=120

(2) Map source attribution: Institute of Geographic Sciences and Natural Resources Research, CAS

(3) Map Terms of Use or License Information

“Data Citation Please state in the data source section: China Multi-year District and County Administrative Boundary Data from the Resource and Environmental Science Data Registration and Publication System, and cite the following data papers:

Xu, Xinliang. Multi-year district and county administrative division boundary data in China. Resource and Environmental Science Data Registration and Publication System (http://www.resdc.cn/DOI), 2023.DOI:10.12078/2023010101”

Question 5.

PLOS requires an ORCID iD for the corresponding author in Editorial Manager on papers submitted after December 6th, 2016. Please ensure that you have an ORCID iD and that it is validated in Editorial Manager. To do this, go to ‘Update my Information’ (in the upper left-hand corner of the main menu), and click on the Fetch/Validate link next to the ORCID field. This will take you to the ORCID site and allow you to create a new iD or authenticate a pre-existing iD in Editorial Manager. Please see the following video for instructions on linking an ORCID iD to your Editorial Manager account: https://www.youtube.com/watch?v=_xcclfuvtxQ

Response 5.

We have registered and linked to ORCID iD as required.

Question 6.

 Please include captions for your Supporting Information files at the end of your manuscript, and update any in-text citations to match accordingly. Please see our Supporting Information guidelines for more information: http://journals.plos.org/plosone/s/supporting-information

Response 6.

Support information file name changed from S1date to S1data. 

Response to "Reviewer #1"：

Question 1. 

WRCC based on the origins of "carrying capacity" concept can have a rigorous physical meaning (e.g., maximum population can live in an area sustainably). Therefore, to address WRCC in this way multiple methods and approaches have been developed. The other way of looking at WRCC and CC is using system of indicators. Although acceptable, this way of estimating WRCC is used mainly for the data scarce situations.

My main comment is why the authors, while having sophisticated input datasets, choose the system of indicators?

For a physical approach of looking at WRCC, please take a look at below references (as a few examples but not limited to):

https://doi.org/10.1016/j.scitotenv.2022.153038

https://doi.org/10.1016/j.landusepol.2010.01.006

https://eprints.qut.edu.au/67485/

https://doi.org/10.1016/j.jclepro.2017.02.110

In my opinion, the manuscript didn't explain enough why the current specific method was chosen, and the text limits the readers just to a group of publications, with high similarity to the current manuscript.

Also, at the end of MS for discussing the results, it doesn't connect the readers to the physics behind the limits on WRCC.

On the other hand, if the system of indicators is still the chosen method, please take a look at alternatives in the following book and explain your choice.

Applied Panarchy: Applications and Diffusion across Disciplines (Page 59 to 68).

Response 1.

The link between the physical concept of carrying capacity and the carrying capacity of water resources is explained below.

Carrying capacity is initially a kinetic physical concept that reflects the maximum carrying capacity of an object before it is destroyed. After the 1960s, anthropologists and biologists applied the concept of carrying capacity in human ecology to describe the maximum tolerance of a regional system to the external environment. The concept and meaning of carrying capacity has since changed from a physical concept to one that reflects the limits of the environment or ecosystem to carry development and specific activities. WRCC can be defined as the ability of a region's resources to support the sustainable development of economic, social and ecological systems under specific conditions.

The explanation of why the indicator system was chosen as a research method is as follows.

The regional scale of the WRCC evaluation study focuses on three main areas: the urban scale, the regional scale, and the watershed scale. The evaluation methods of WRCC are roughly divided into two kinds: one method is to construct the index system from the appearance of water resources carrying capacity, and to derive the comprehensive index value through mathematical methods, and to calculate the scoring value of the target object in different time and space, which includes fuzzy comprehensive judgement method, principal component analysis method, and set-pair analysis method. Another approach is to start from the internal role of the water resources carrying capacity system, construct mathematical equations to simulate the development of various factors, and couple the equations into a quantitative model of carrying capacity through interconnections, so as to simulate the maximum carrying capacity of water resources, which mainly includes the conventional trend method, the state-space method, the system dynamics method, the projection tracing method, the ecological footprint method, and so on. In this study, due to the lack of data related to the capacity of the water environment, the capacity of the water environment to accommodate pollution, the amount of permissible pollution loads in the water environment, and only have the basic data related to the economy, society, resources, the environment, and ecology, so we start from the surface to build an indicator system evaluation, and use the indicator system approach to the evaluation of the water resources carrying capacity.

Question 2. 

There is a high level of math and system of equations for the methods and analysis. However, I think the main objective is lost. The main goal is to assess the capacity of the land based on its resources. For a successful use of equations, I think the steps need more clear interpretations. For example, take a look at these two examples:

https://doi.org/10.1126/science.7618100

https://doi.org/10.1016/j.heliyon.2023.e15079

Response 2.

WRCC is a comprehensive indicator that involves the coupling of multiple systems, including water resources, population, ecology and socio-economics. 

Set-pair analysis reflects the complex systematic structure of the set of pairs of water resources carrying capacity assessment samples and assessment criteria, as it can analyze and quantitatively express the degree of conformity between assessment samples and assessment criteria in an insightful and comprehensive way. In this study, in order to evaluate the state of water resources carrying capacity and the main obstacle factors, the entropy weighting method was used to assign weights to the evaluation indicators to avoid the interference of subjective factors, the set-pair analysis method was used to evaluate the state of water resources carrying capacity, and the obstacle degree model was used to diagnose the main obstacle factors.

Question 3. 

The PSR, is one of the analytical framework to assess a complex SES systems. In a few sentences explain the existence of others. Also, the more comprehensive way is to use DPSIR framework. Please justify why the MS narrow it down to PSR framework. The following reference can help

https://www.jstor.org/stable/26269404

Response 3.

The PSR model was chosen for the following reasons.

D is the driving force and I is the impact in the DPSIR model.

Driving force generally refers to the changes of the external objective existence and its impact on water resources, including natural driving force and socio-economic driving force, of which the natural driving force includes two kinds of climate change and natural disasters. For the urban agglomeration in central Yunnan, several cities are close to the same natural driving force due to their geographic proximity, so there is no need for comparative study, while the socio-economic driving force mainly comes from human activities, which is closer to the meaning of pressure in the model. Therefore, for the study of water resources carrying capacity of central Yunnan urban agglomeration, the driving force indicators are included in the pressure subsystem indicators. 

The impact generally refers to the impact of the state of the water resources carrying capacity on urban development. Since several cities are concentrated and there is not much difference in the natural environment, the impact on urban development is generally considered from the social and economic aspects, whereas the state indicators are selected to refer to the state change of the water resources system in the three aspects of the social, economic, and ecological aspects, and the coordinated development of the states is also taken into consideration. Therefore, the state subsystem indicator includes the impact indicator.

Question 4. 

Please use the standard and acceptable terminology for the concepts. For example "overload" in Line 287 should be replaced with "overshoot". Or, "the state of carrying" in Line 284 is not a correct term. Please rephrase. Similar examples exist in the MS (Lines 311, 325, 333 and multiple other places).

Response 4.

We have checked the relevant expressions in the manuscript and completed the revisions.

Question 5. 

In Lines 258-260 the MS says "water-saving technologies in agriculture, adopting sprinkler irrigation, tube irrigation and other irrigation methods."

This argument is not true. The irrigation technologies can save water at the farming scale (small scale). However, at the watershed scale and for the large scale planning, the prom

---

## [Decision Letter · Decision Letter 1]

25 Jul 2024

Study on water resources carrying capacity based on Pressure-State-Response modeling：An empirical study of the urban agglomeration in Central Yunnan, China

PONE-D-24-09140R1

Dear Dr. Wang,

We’re pleased to inform you that your manuscript has been judged scientifically suitable for publication and will be formally accepted for publication once it meets all outstanding technical requirements.

Kind regards,

Sivasankar Koppala

Academic Editor

PLOS ONE

Additional Editor Comments (optional):

Reviewers' comments:

Reviewer's Responses to Questions

**Comments to the Author**

1. If the authors have adequately addressed your comments raised in a previous round of review and you feel that this manuscript is now acceptable for publication, you may indicate that here to bypass the “Comments to the Author” section, enter your conflict of interest statement in the “Confidential to Editor” section, and submit your "Accept" recommendation.

Reviewer #2: All comments have been addressed

2. Is the manuscript technically sound, and do the data support the conclusions?

Reviewer #2: Yes

3. Has the statistical analysis been performed appropriately and rigorously? 

Reviewer #2: Yes

4. Have the authors made all data underlying the findings in their manuscript fully available?

Reviewer #2: Yes

5. Is the manuscript presented in an intelligible fashion and written in standard English?

Reviewer #2: Yes

6. Review Comments to the Author

Reviewer #2: The manuscript evaluates the WRCC of the UACY in China's plateau region, taking into account economic, social, resource, environmental and ecological factors, and explores the temporal and spatial evolution of the WRCC of the UACY, which provides data support for the formulation of water resources related policies and the rational scheduling of water resources in the plateau region. The manuscript has innovative and practical meaningful. It is recommended that this paper should be accepted for publication.

7. PLOS authors have the option to publish the peer review history of their article (what does this mean?). If published, this will include your full peer review and any attached files.

Reviewer #2: No

---

## [Editor Report · Acceptance letter]

29 Jul 2024

PONE-D-24-09140R1 

PLOS ONE

Dear Dr. Wang, 

I'm pleased to inform you that your manuscript has been deemed suitable for publication in PLOS ONE. Congratulations! Your manuscript is now being handed over to our production team.

Kind regards, 

on behalf of

Dr. Sivasankar Koppala 

Academic Editor

PLOS ONE